# Changing risk factors for developing SARS-CoV-2 infection from Delta to Omicron

**Paul R. Hunter** [ID], **Julii Brainard** [ID] *

Norwich Medical School, University of East Anglia, Norwich, United Kingdom

* J.Brainard@uea.ac.uk

## Abstract

### Background

One of the few studies to estimate infection risk with SARS-CoV-2 in the general population was the UK Office of National Statistics Infection Survey. This survey provided data that allowed us to describe and interpret apparent risk factors for testing positive for SARS-CoV-2 in a period when variants and COVID-19 controls experienced large changes.

### Method

The ONS published estimates of likelihood of individuals testing positive in two week monitoring periods between 21st November 2021 and 7th May 2022, relating this positivity to social and behavioural factors. We applied meta-regression to these estimates of likelihood of testing positive to determine whether the monitored potential risk factors remained constant during the pandemic.

### Results

Some risk factors had consistent relationship with risk of infection (always protective or always linked to higher risk, throughout monitoring period). Other risk factors had variable relationship with risk of infection, with changes seeming to especially correlate with the emergence of Omicron BA.2 dominance. These variable factors were mask-wearing habits, history of foreign travel, household size, working status (retired or not) and contact with children or persons age over 70.

### Conclusion

Relevance of some risk factors to likelihood of testing positive for SARS-CoV-2 may relate to reinfection risk, variant infectiousness and status of social distancing regulations.

## Introduction

Since the start of the COVID-19 pandemic in December 2019 there have been hundreds if not thousands of publications seeking to address the risk factors for deleterious COVID-19

**Data Availability Statement:** The data used in this study are not owned by the authors and therefore cannot be directly publicly redistributed by the authors for instance, by placing the dataset in a public archive. However, the data are currently

publicly available at: https://www.ons.gov.uk/peoplepopulationandcommunity/healthandsocialcare/conditionsanddiseases/datasets/coronaviruscovid19infectionsinthecommunityinengland/25may2022.

**Funding:** PRH and JB were funded by the National Institute for Health Research Health Protection Research Unit (NIHR HPRU, grant NIHR200890) in Emergency Preparedness and Response at King's College London in partnership with the UK Health Security Agency (UKHSA), in collaboration with the University of East Anglia. The views expressed are those of the authors and not necessarily those of the NHS, the NIHR, any of our employers, the Department of Health or the UKHSA. The funders had no role in study design, data collection and analysis, decision to publish, or preparation of the manuscript.

**Competing interests:** The authors have declared that no competing interests exist.

outcomes. However, relatively few of those publications were well designed to identify risk factors for infection, whether or not symptomatic, rather than risk factors for symptomatic infection, hospitalisation and deaths. One of the more powerful studies was based on the UK Biobank cohort which analysed a cohort of almost a quarter of a million people. In that study, non-modifiable risk factors for confirmed infection in March to May 2020 (when testing in the UK was limited to persons with substantial medical need or occupational risk) included male sex, black ethnicity and socioeconomic deprivation [1]. Another comprehensive government agency early review of the evidence into disparities in risk and outcomes from COVID-19 infections in patients detected between February and April 2020 inclusive [2] found that age and ethnicity were major drivers of infection and severe outcome. These early pandemic studies and many other COVID-19 risk studies overwhelmingly related to patients with symptomatic illness if not actual severe disease.

As far as we are aware, the only community-based studies of infection as opposed to symptomatic infection that also reported on behavioural risk factors is the coronavirus infection survey in the UK conducted by the UK Office of National Statistics (CIS ONS). This was a cohort study that aimed to sample up to 150,000 individuals over the age of 2 years every fortnight [3]. The CIS ONS was the least biased epidemic tracker because it was designed to collect data on all cases, including very mild, pre-symptomatic and asymptomatic cases [4]. As such, the CIS ONS gave the most complete picture of infection spread at any one time, and risk of infection, at any severity.

ONS published its estimates of prevalence and incidence of COVID-19 weekly, the last report was in March 2023 (ONS 2023b). In addition, ONS published risk factor analyses in its "characteristics report" once or twice a month from June 2021 to November 2022 [5]. These analyses include estimates of risk by during two-week periods for amongst other things prior vaccination and infection, occupational factors, age and gender, overseas travel, and wearing of face coverings. Risk factors were presented for each of subsequent two-week periods enabling determination of association between specific factors and case status.

We report a study to determine whether risk factors for infection changed after the emergence of the omicron variant. We explored whether risk factors for infection remained the same as they were prior to the emergence of Omicron or if they changed. In addition, we wanted to know whether risk avoidance behaviours, such as mask wearing, continued to be associated with reduced risk.

## Methods

All data included in this study come from publicly available analyses produced by the UK Office of National Statistics (ONS) and were published online [5]. The ONS covid survey recruited about 200,000 people and took throat swabs for COVID-19 using rt PCR every two weeks. Basic prevalence estimates were published every week and for a limited time period, the ONS published its COVID-19 characteristics analyses where they reported risk factors for testing positive in each two-week period. The outcome variable in the ONS analyses was whether or not someone tested positive and did not distinguish between whether someone had symptoms or not. The data for the analyses presented here were published on 25th May 2022 [5]. After that date, analyses changed to presenting data on risk factors for reinfection only. The analyses used here presented estimated likelihood of testing positive along with the standard error for various risk factors in sequential fortnightly periods between 21st November 2021 and 7th May 2022, as this was the period when ONS published the relevant analyses. These likelihoods were calculated by ONS from models that controlled for age, region, sex, ethnicity, deprivation, household size, multi-generational household, and urban/rural classification as

well as vaccination history and history of prior infection. The UK Census as undertaken by ONS defines a mutigenerational household as "*Households where people from across more than two generations of the same family live together. This includes households with grandparents and grandchildren whether or not the intervening generation also live in the household*" [6].

The primary data is not publicly available, and our analyses were done on the published summary data for each fortnightly time period. We consider separately the likelihood of testing positive over the whole period, as well as the likelihood of testing positive in any single report period (n = 12 in the monitoring period). Because we do not have access to the primary data, we used meta-regression, with the metareg tool in STATA 17.0. Initial analyses were the random effects pooled likelihood of testing positive for each risk factor over the included time periods. Then an analysis for trend was undertaken by adding period number (one to 12) as a fixed effect to the model. Those risk factors with a significant trend analysis were then used in fir tree graphs generated using STATSDirect 3.6. Significance threshold was set at p < 0.05.

We discuss the results with respect to concurrent dominant variant and government epidemic control policies. The time period covered by these analyses include the final few weeks when the Delta variant was dominant in the UK (until 19 December 2021), after which Omicron variant BA.1 was dominant until 1 March 2022 and Omicron variant BA.2 was dominant until 15 June 2022 [7]. The introduction and easing of non-pharmaceutical control measures varied between the four devolved administrations that comprise the UK, but the data presented here were most influenced by concurrent controls regime in England (the most populous administrative region). The end of all COVID-19 controls in England was announced on 21 Feb 2022 [8], which stated that the legal requirement to self-isolate stopped on 24 Feb and that guidelines to self-isolate as well as free testing would cease on 1 April 2022. Because this is a secondary analysis of published aggregate data, ethical approval was not required.

## Results

The meta-regression analyses are shown in Table 1. Likelihoods of testing positive allowing for selected risk factors in specific periods are shown in Figs 1–4. S1-S29 Figs in S1 Appendix show the pattern for those risk factors for which trend was not statistically significant at the p = 0.05 level. In this monitoring period, the risk factors not associated with any likelihood of testing positive and for which there was no significant trend were work in health or social care (p = 0.848), work in a patient facing role (p = 0.534), work in a care or nursing home (p = 0.468) and work in contact with others (p = 0.066). Variables that were associated with difference in the likelihood of testing positive but there was no significant change (p > 0.005 for trend over time) over the study period were:

- Gender, with females consistently less likely to test positive (likelihood -0.051, p < 0.001)

- Being an ethnic minority was linked to less positivity (likelihood -0.119, p = 0.012)

- People from a multi-generational household were less likely to test positive (likelihood -0.055, p = 0.025)

- People were less likely to test positive if in the previous 28 days they had contact with a care home (likelihood –0.126, p = 0.008) or a hospital (likelihood -0.176, p < 0.001)

- Tobacco smokers were less likely to test positive (likelihood -0.170, p < 0.001)

- Living in a house with someone who was disabled was associated with lower risk (likelihood –0.126, p < 0.001)

- Being disabled meant lower risk of having infection (likelihood -0.075, p = 0.005)

**Table 1. Risk factors for testing positive.**

| Predictor variable | | likelihood of testing positive | Standard error | P value | t for trend over time | P for trend over time |
|---|---|---|---|---|---|---|
| Gender | Male | 0 | | | | |
| | Female | -0.051 | 0.01 | <0.001* | -0.4 | 0.700 |
| Household size | 1 | 0 | | | | |
| | 2 | 0.128 | 0.023 | <0.001* | 3.12 | 0.011* |
| | 3 | 0.207 | 0.033 | <0.001* | 0.33 | 0.745 |
| | 4 | 0.263 | 0.044 | <0.001* | -1.41 | 0.190 |
| | 5+ | 0.282 | 0.055 | <0.001* | -2.22 | 0.036* |
| Multi-generational household | No | 0 | | | | |
| | Yes | -0.055 | 0.021 | 0.025* | -0.55 | 0.596 |
| Ethnic minority | No | 0 | | | | |
| | Yes | -0.119 | 0.04 | 0.012* | -0.12 | 0.904 |
| Rurality | Major urban area | 0 | | | | |
| | Urban city or town | 0.012 | 0.021 | 0.588 | 3.61 | 0.005* |
| | Rural town | -0.034 | 0.038 | 0.387 | 1.92 | 0.084 |
| | Rural village | -0.078 | 0.047 | 0.122 | 1.44 | 0.181 |
| Employment status | Employed working | 0 | | | | |
| | Employed, not working | 0.05 | 0.042 | 0.254 | -0.85 | 0.416 |
| | Not working | -0.199 | 0.039 | <0.001* | 2.12 | 0.060 |
| | Child/student | -0.165 | 0.055 | 0.012* | 1.97 | 0.077 |
| | Retired | -0.127 | 0.029 | 0.001* | 4.65 | 0.001* |
| Work in health or social care | No | 0 | | | | |
| | Yes | -0.011 | 0.057 | 0.848 | 1.96 | 0.079 |
| Work in patient facing role | No | 0 | | | | |
| | Yes | 0.033 | 0.052 | 0.534 | 1.69 | 0.122 |
| Work in a care or nursing home | No | 0 | | | | |
| | Yes | 0.047 | 0.063 | 0.468 | 1.44 | 0.181 |
| Work in contact with others | No | 0 | | | | |
| | Yes | 0.052 | 0.026 | 0.066 | -0.82 | 0.430 |
| Contact with care homes in the last 28 days | No one in house | 0 | | | | |
| | No, but someone else in my household has | -0.048 | 0.032 | 0.162 | 0.97 | 0.356 |
| | Yes | -0.126 | 0.039 | 0.008* | 1.77 | 0.106 |
| Contact with hospitals in the last 28 days | No one in house | 0 | | | | |
| | No, but someone else in my household has | -0.1 | 0.014 | <0.001* | 0.14 | 0.891 |
| | Yes | -0.176 | 0.013 | <0.001* | 1.17 | 0.270 |
| Smoking status | Non | 0 | | | | |
| | Only vape | -0.076 | 0.041 | 0.089 | -1.42 | 0.186 |
| | Tobacco smoker | -0.17 | 0.033 | <0.001* | -2.1 | 0.062 |
| Disabled | No | 0 | | | | |
| | Yes | -0.075 | 0.022 | 0.005* | 2.1 | 0.063 |
| Travelled abroad in past 28 days | No | 0 | | | | |
| | Yes | 0.275 | 0.052 | <0.001* | 2.41 | 0.037* |
| Adults, living in a household with someone aged 16 or under | No | 0 | | | | |
| | Yes | -0.035 | 0.051 | 0.504 | -2.92 | 0.015* |
| Aged under 70 years, living in a household with someone aged 70 or over | No | 0 | | | | |
| | Yes | -0.097 | 0.040 | 0.034* | 2.74 | 0.021* |

*(Continued)*

**Table 1.** (Continued)

| Predictor variable | | | likelihood of testing positive | Standard error | P value | t for trend over time | P for trend over time |
|---|---|---|---|---|---|---|---|
| Someone else in household is disabled | No | | 0 | | | | |
| | Yes | | -0.126 | 0.018 | <0.001* | 1.86 | 0.092 |
| School aged children wear face coverings in enclosed spaces | Always | | 0 | | | | |
| | Sometimes | | 0.014 | 0.049 | 0.783 | -2.12 | 0.060 |
| | Never | | 0.028 | 0.043 | 0.533 | -2.32 | 0.043* |
| Adults—use of face coverings in enclosed spaces | Always | | 0 | | | | |
| | Sometimes | | 0.089 | 0.019 | 0.001* | -1.55 | 0.152 |
| | Never | | 0.146 | 0.053 | 0.019* | -2.79 | 0.019* |
| School aged children—use of face coverings in school or work | Always | | 0 | | | | |
| | Sometimes | | -0.089 | 0.061 | 0.071 | -1.33 | 0.212 |
| | Never | | 0.051 | 0.067 | 0.462 | -2.59 | 0.027* |
| Adults—use of face coverings in school or work | Always | | 0 | | | | |
| | Sometimes | | 0.057 | 0.029 | 0.072 | -1.46 | 0.174 |
| | Never | | 0.077 | 0.028 | 0.019* | -2.97 | 0.014* |
| Total physical contacts in the last 7 days | 0 | | 0 | | | | |
| | 1 to 10 | | 0.155 | 0.014 | <0.001* | 0.46 | 0.653 |
| | 11 to 20 | | 0.255 | 0.015 | <0.001* | -0.73 | 0.481 |
| | 21 to 40 | | 0.253 | 0.025 | <0.001* | -0.68 | 0.512 |
| | >40 | | 0.237 | 0.026 | <0.001* | -0.72 | 0.485 |
| Total socially distanced contacts in the last 7 days | 0 | | 0 | | | | |
| | 1 to 10 | | 0.043 | 0.184 | 0.038 | 1.66 | 0.128 |
| | 11 to 20 | | 0.161 | 0.023 | <0.001* | 1.46 | 0.175 |
| | 21 to 40 | | 0.208 | 0.022 | <0.001* | 1.62 | 0.130 |
| | >40 | | 0.203 | 0.024 | <0.001* | 1.62 | 0.137 |

Note

*: significant p-value. Significance threshold was set at p < 0.05.

- Increasing numbers of physical contacts and socially distanced contacts in the previous 7 days were also associated with increased risk. For physical contacts risk plateaued about 10 such contacts and for socially distanced contacts these plateaued about 20 such contacts.

Other risk factors for infection were not as simple to describe and were even variable during the monitoring period. There was a complex relationship between household size and positivity. People living with others, especially two or more others, had consistent increased risk of testing positive (Table 1, likelihood of testing positive increased with household size, p always < 0.001). Figs 1–4 show the likelihood of testing positive for selected risk factors with confidence intervals, in each of the 12 two week periods. Between the figures on each row in the figures is a chart indicating changing proportions of tested samples in each of the Delta and Omicron variants over the concurrent time period, earliest at top to latest at bottom. This may facilitate interpretation about how changes in variants changed or not with each risk factor. Fig 1 focuses on household size, with likelihoods shown compared to singleton households. The likelihoods of testing positive generally decreased for households with 3 or more members after 12 Feb 2022, but the likelihood of testing positive increased for households with size = 2 in the period from January to May 2022. The greatest decrease in likelihood of testing positive was for the largest households (n = 5+). By the end of the study period people in larger

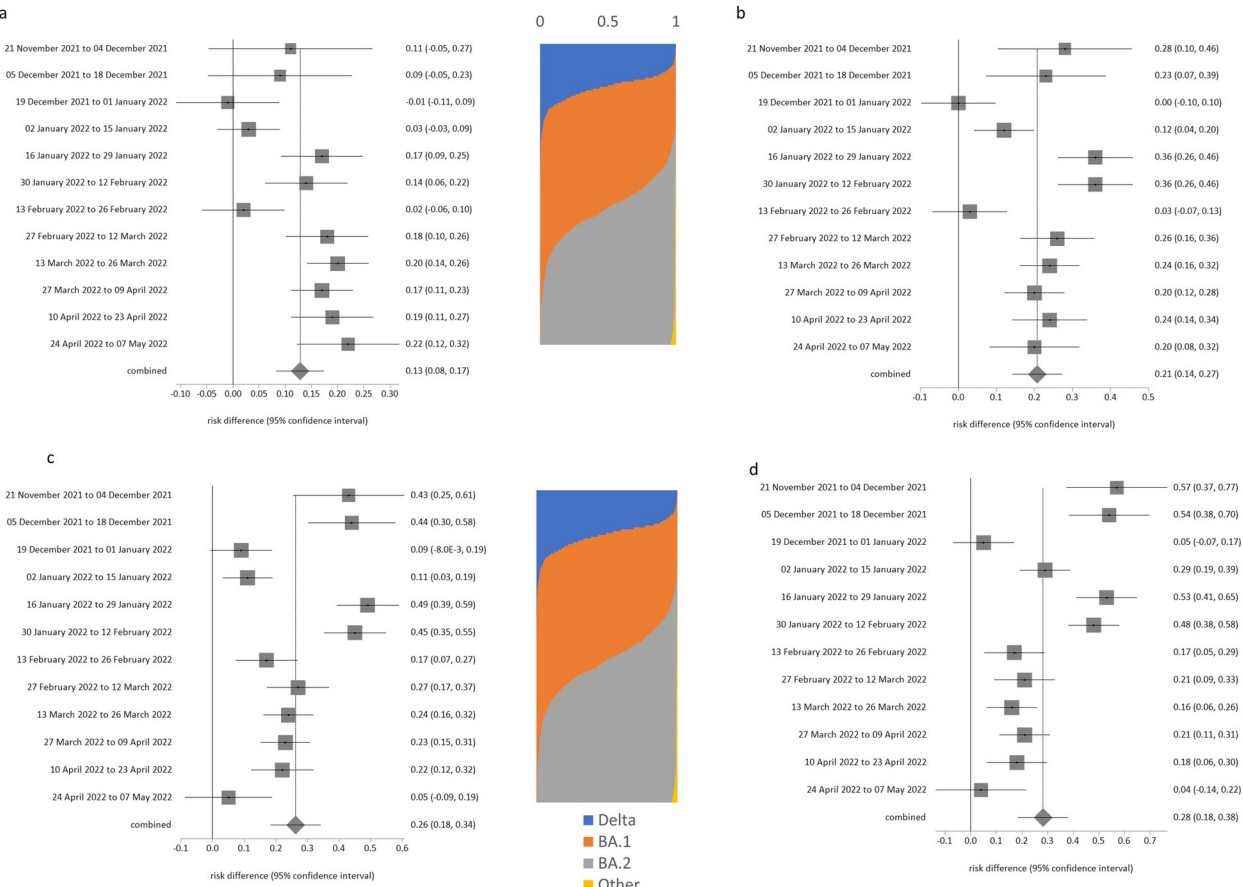

**Fig 1. Risk of testing positive for SARS-CoV-2 by household size compared to households with just a single person.** a. 2 people in household, b. 3 people in household, c. 4 people in household, d. 5 or more people in household.

households (4 and above) had negligibly greater risk than people living in singleton households.

Although overall there was no difference in risk between people who lived with or did not live with someone aged ≤ 16 years (likelihood = 0.035, p = 0.504), there was evidence of significant change (trend over time p = 0.015). Fig 2A shows that there was a notable negative association with risk from about end February onwards. People aged under 70 years who lived with someone 70 years or older had overall lower likelihood of testing positive (likelihood = -0.097, p = 0.034), and this relationship also had a significant change over time (p = 0.021). However, detailed analysis (Fig 2B) is that those sharing households with older adults were actually only at reduced risk for the first half of the study period. By about mid-February there was little effect on risk of infection.

Compared to always wearing face coverings, school-aged children who never wore face coverings had an insignificantly higher likelihood of testing positive (Table 1: likelihood was 0.028 in enclosed spaces, p = 0.533; likelihood was 0.051 at school, p = 0.462). However, both of these comparisons had significance for trend over time. Fig 3A and 3B show that the likelihood of testing positive for never-wearers was higher (significantly so) in the early monitoring period (before end February 2022) in both settings, and much lower in the March-May period (although not usually significantly different from the always-wearers at p < 0.05).

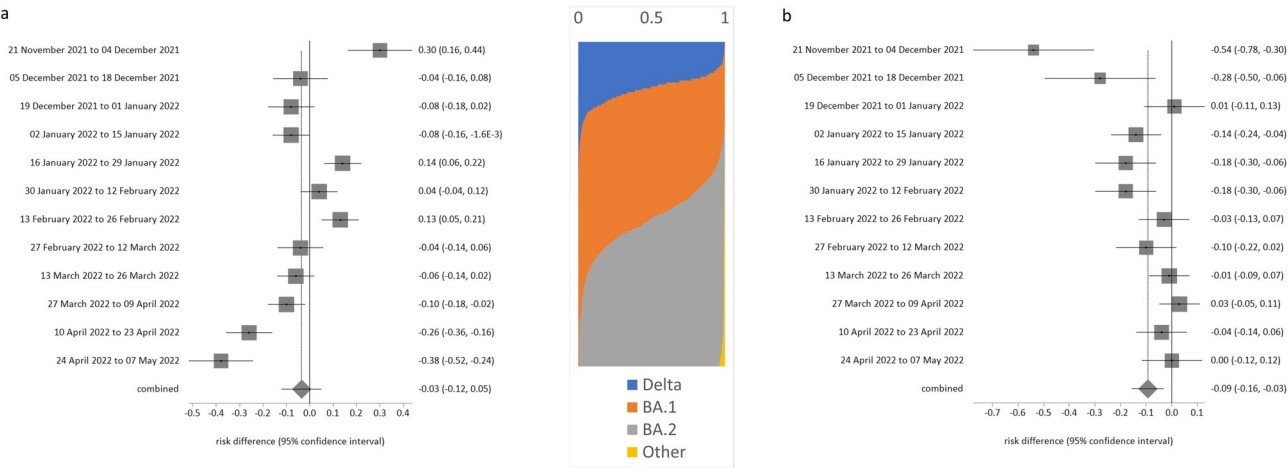

**Fig 2. Risk of testing positive for SARS-CoV-2, with respect to ages of other household members.** a. Adults living with someone age 16 years or younger, b. Adults aged under 70 years and living with someone age 70 years or older.

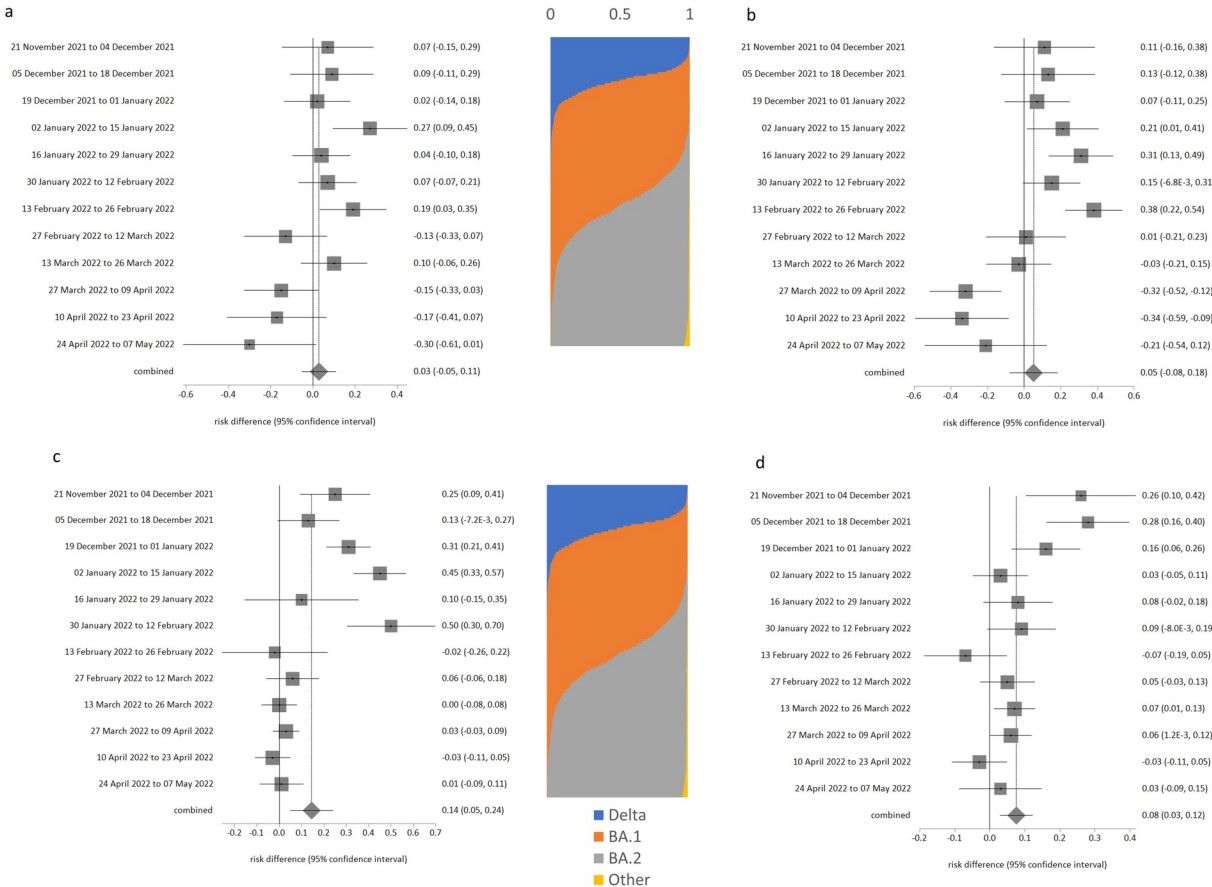

**Fig 3. Risk of testing positive for SARS-CoV-2 when never wearing a face covering, compared to always wearing one.** a. School children in enclosed spaces, b. School children in school, c. Adults in enclosed spaces, d. Adults at school or at work.

Compared to always wearing face-coverings, adults had higher risk of testing positive (Table 1: 0.146 in enclosed spaces, p = 0.019; 0.077 at school or work, p = 0.019). Again, both of these comparisons had significance for trend over time. Fig 4C and 4D show that never wearing a face covering was much more strongly linked to positivity before early-mid January, and by end May not wearing a face covering in adults had no effect on risk for adults in these settings.

There was no evidence for trend over time for likelihood of infection in rural towns or villages (p > 0.005 in Table 1), but change over time was indicated for minor urban areas compared to major urban areas (p = 0.005). Compared to living in a major urban area, people in smaller urban locations had lower risk of testing positive during the first half of the monitored period (Fig 4A), and then greater risk of positivity which coincided with the emergence of BA.2 dominance and announcements about lifting of all COVID-19 restrictions on or around 1 March 2022.

Being retired was associated with reduced risk (Table 1, likelihood -0.127, p < 0.001) compared to those in work overall, but any protective effect had disappeared by 27 Feb 2022 (Fig 4B), which coincided with the start of the second Omicron wave.

Travel abroad was not associated any increased risk before 2 January, but after Omicron became dominant, travel abroad had a substantially increased association with risk of infection (Fig 4C), likelihood = 0.45 or 0.54 in January 2022. This elevated likelihood of testing positive linked to travel abroad subsequently declined somewhat in the Feb-May 2022 period.

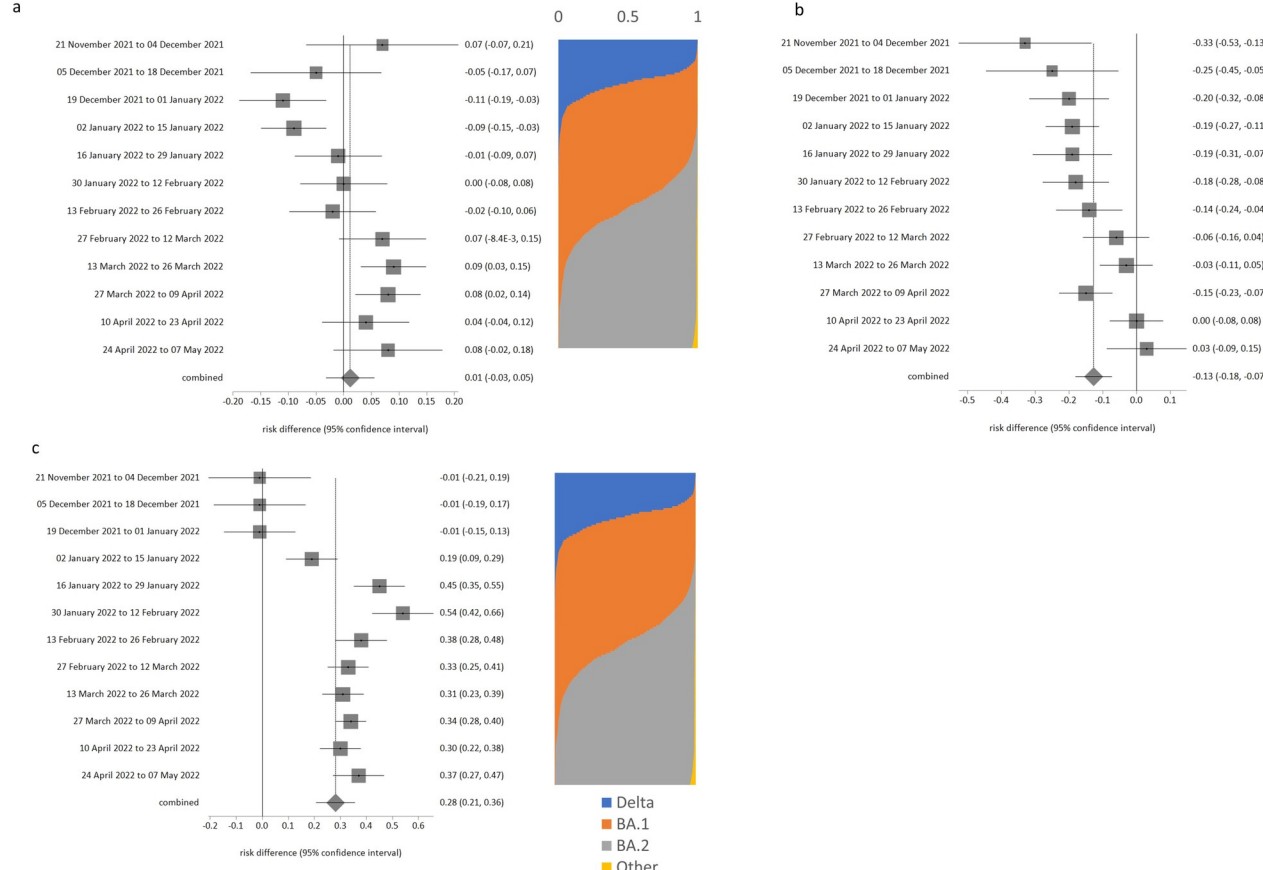

**Fig 4. Risk of testing positive by rurality, retirement status or recent travel history.** a. Living in urban city or town compared to major urban area, b. Being retired compared to being employed and going to work, c. Travelled abroad in recent 28 days compared to no foreign travel in last 28 days.

## Discussion

We have shown that effect of many of the posited risk or protective factors for COVID-19 infection varied during the course of the pandemic in the UK. Some factors that were associated with lower risk for testing positive during the Delta and Omicron BA.1 periods (eg., such as being retired or wearing face coverings) were no longer associated with any risk difference or even an increased risk during the second Omicron wave and after it was announced that restrictions would be lifted. Some factors that were associated in our data with increased risk of testing positive early on, such as living with someone under age 16 or children wearing face-coverings, were associated with much reduced or even no risk later on. Multi-generational households had no time trend change in risk of infection, although size of household did have a changing relationship with risk of infection over time; this suggests that multi-generational households are not reliably also the largest households. Some risk factors such as working in health or social care or in contact with others, were often found to be important in the first year of the pandemic [by July 2020: 9, 10], but were not associated with overall higher or changing risk of infection in our study period. Being an ethnic minority was strongly associated with increased risk in the first few months of the UK epidemic [by May 2020: 11], but was associated with lower risk and no significant trend change during our full monitoring period.

We are not the first investigators to find that risk factors for COVID-19 changed during the course of a within-country epidemic. In a large case-controlled study of symptomatic people with confirmed SARS-CoV-2 positivity in the USA, Hansen and colleagues (2022) [12] noted that reported contact with a known case was highly associated with infection during the Delta period but much less so for Omicron. Unlike our analysis, that study did not include any behavioural characteristics. In a study using international datasets, Sharma and colleagues (2021) [13] found that many of the non-pharmaceutical interventions that were effective in early 2020, such as school closures and gathering bans, were less effective in the winter 2020/21 wave.

The finding that wearing face coverings ceased being protective after the first Omicron wave is worthy of further discussion. It is possible that behavioural changes drove this change in part, as wearing face coverings became less expected after the announcements that COVID-19 restrictions were being lifted. Physical contact rates in all settings likely rose sharply in wake of the announcements about the plans for "Living with COVID-19". However, behavioural changes may not explain all changes in risk factor association with infection. Biological changes such as increased infectivity with new variants or waxing and waning immunity in the population seem likely to also cause differences in risk or protective factors. The balance of evidence is that wearing face coverings reduces transmission of respiratory infections in community settings and did reduce transmission of COVID-19 (Chou et al. 2023). The question, however, is by how much. Systematic review of pre-pandemic evidence [14] and analysis of original survey data during the COVID-19 pandemic [15] both indicated that mask wearing could or did reduce transmission of SARS-CoV-2 by about 19%. But these conclusions were derived mainly from data prior to the emergence of Omicron variants.

From the data presented here, prior to Omicron BA.2, never wearing a mask was associated with an increased risk of around 30% in adults and 10% in children. But by the second Omicron wave (mid to late February 2022 onwards) there was no protective effect from mask wearing in adults and possibly an increased risk of infection in children. The need for children to wear masks in school has been a topic of considerable debate. Budzyn and colleagues [16] reported that in the summer of 2021, those US counties that did not have school mask mandates in place reported a greater increase in paediatric COVID-19 infections compared to those that did. However, this study only covered the first two weeks of the school term. In an

analysis of the same data source but including data from many more counties and for a longer period of time. Chandra and Høeg (2022) [17] found that by the ninth week after school start, any association between mask mandates in schools and reduced risk of infection had disappeared. They also concluded that once confounding factors such as poverty and a Social Vulnerability Index were included, counties imposing mask mandates saw more paediatric infections.

The question remains what could be the mechanism(s) that help to explain why protective interventions that were associated with decreased risk early in the pandemic become less protective or even associated with increased risk. Similarly, we can ask why did some variables that were associated with increased risk early in the pandemic cease to be associated with risk of infection later.

We propose that two biological mechanisms were relevant. The first hypothesis is that because Omicron was more infectious than previous variants [18], interventions that had previously been able to suppress the effective reproduction number to keep it close to 1.0 were no longer as effective, thereby increasing risk of infection in circumstances where previously risk was relatively low. Our finding that the risk associated with international travel increased substantially after the appearance of Omicron supports this hypothesis. However, we note that the change in risk associated with foreign travel applies to a period when international travel was rapidly becoming easier (pandemic travel restrictions were being eased). This change in opportunity may have correlated with changes in other risk factors and behaviours not recorded. In any event, the first hypothesis does not seem likely to explain all of our findings, such as the shift from reduced to increased risk associated with wearing face coverings in school.

The second hypothesis is linked to immunity from prior infection. People who accept greater risk are more likely to have been infected and so more likely to be immune later in the pandemic and at lower risk of infection. Early in the pandemic most published epidemic models of the COVID-19 pandemic were of the SIR or SEIR form (Susceptible, Exposed, Infected and Recovered) [19]. However, these models perform badly after the first wave for infections like COVID-19 where immunity to reinfection is not long lasting [20]. As is now well known, COVID-19 infection and immunisation only provides immunity against further infection for a relatively limited period, of the order of just a few months [21]. A more appropriate epidemic model for COVID-19 at least after the first wave is the SEIRS model (Susceptible, Exposed, Infected, Recovered and Susceptible) [20]. In the SEIRS model, an epidemic in a previously naive population will pass through a series of diminishing waves until the infection becomes endemic. The COVID-19 pandemic has generally followed this path [22]. As the epidemic transitions to become endemic the drivers of infection change and then a major driver of infection rates is the rate at which immunity is lost. At this point, interventions that would have had a major impact on slowing transmission early in the epidemic phase have less influence. That non pharmaceutical interventions have value primarily in the early stages of a pandemic is something that has been known for some time. Their value lies largely in delaying most people's infections until a suitable vaccine becomes available [23].

Ideally, we would have used the primary individual level data and undertaken some form of longitudinal panel or multilevel mixed effects regression analysis. However, we only had access to summary data for each time period. Thus, the data were effectively clustered by time period [24]. Calculating summary data for each cluster followed by standard analysis of that summary data is an acceptable approach [23]. The main weakness of this approach is that just using the central estimate does not take account of the precision of each summary value [23]. Meta-analysis is an approach that combines the results from different studies whilst accounting for the

precision of each study [25]. Thus, our use of meta-regression allowed us to use the summary data from each time period whilst adjusting for the precision of each estimate.

In conclusion we have shown that risk factors for infection and effectiveness of public health interventions are not fixed but sometimes, and may commonly, change during the course of a pandemic. It is plausible and seems most likely that at least partial explanations for such changes are increased immunity in people at high risk of infection as the epidemic progressed, and the rate at which such immunity is lost or maintained by repeat exposure.

## Supporting information

**S1 Appendix. Changing covid risk factors.**
(PDF)

## Acknowledgments

We acknowledge the essential contribution from the Office of National Statistics.

## Author Contributions

**Conceptualization:** Paul R. Hunter.

**Data curation:** Paul R. Hunter.

**Formal analysis:** Paul R. Hunter.

**Funding acquisition:** Paul R. Hunter.

**Investigation:** Paul R. Hunter.

**Methodology:** Paul R. Hunter.

**Supervision:** Paul R. Hunter.

**Visualization:** Paul R. Hunter.

**Writing – original draft:** Paul R. Hunter, Julii Brainard.

**Writing – review & editing:** Paul R. Hunter, Julii Brainard.

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
