## [Decision Letter · Decision Letter 0]

12 Dec 2023

PONE-D-23-35350Changing risk factors for developing SARS-CoV-2 infection from Delta to OmicronPLOS ONE

Dear Dr. Brainard,

Thank you for submitting your manuscript to PLOS ONE. After careful consideration, we feel that it has merit but does not fully meet PLOS ONE’s publication criteria as it currently stands. Therefore, we invite you to submit a revised version of the manuscript that addresses the points raised during the review process.

We look forward to receiving your revised manuscript.

Kind regards,

Marwan Osman

Academic Editor

PLOS ONE

Journal Requirements:

"PRH and JB were funded by the National Institute for Health Research Health Protection Research Unit (NIHR HPRU, grant NIHR200890) in Emergency Preparedness and Response at King’s College London in partnership with the UK Health Security Agency (UKHSA), in collaboration with the University of East Anglia. The views expressed are those of the authors and not necessarily those of the NHS, the NIHR, any of our employers, the Department of Health or the UKHSA."

Reviewers' comments:

Reviewer's Responses to Questions

**Comments to the Author**

1. Is the manuscript technically sound, and do the data support the conclusions?

Reviewer #1: Partly

Reviewer #2: Partly

2. Has the statistical analysis been performed appropriately and rigorously? 

Reviewer #1: Yes

Reviewer #2: Yes

3. Have the authors made all data underlying the findings in their manuscript fully available?

Reviewer #1: Yes

Reviewer #2: Yes

4. Is the manuscript presented in an intelligible fashion and written in standard English?

Reviewer #1: Yes

Reviewer #2: Yes

5. Review Comments to the Author

Reviewer #1: 1. The method section in the Abstract is too short. Please add more information on the data and the statistical analyses. The estimates should be reported in the result section. The conclusions should be the implications and interpretation of the results.

2. I suggest replacing the term “likelihood of testing positive” with the risk/likelihood of infection.

3. As the study period across phases where different variants are circulating, a variable related to these distinct phases should be included in the meta-regression model instead of the period number, such that the interpretation of the estimates can be compared among different variants' predominant phases.

4. Given that the meta-regression was often used in the context of substantial between-study variations, the metric I2 should be reported for each pooled estimate.

5. Please be consistent about the term “Covid-19” (e.g., instead of “covid”) throughout the text.

6. Why are the results for multigeneration households reversed compared to household sizes? Please state the difference between multigeneration household and normal household.

7. Please change Table 1 to a figure similar to figure 1-4 but combining all the pooled estimates shown in Table 1.

8. It is unclear why the authors only show the detailed estimates during the periods for 4 strata in table1. I suggest generating the same figure for all other strata as well and moving all these figures to the Appendix.

9. The strata “Adults, living in a household with someone aged 16 or under” and “Aged under 70 years, living in a household with someone aged 70 or over ” are different from the figure2 legend.

10. Please use add the full form of “effective R”.

Reviewer #2: The authors analyzed SARS-CoV-2 infection risk in the general population using the UK Office of National Statistics Infection Survey. They used meta-regression of likelihood estimates during the pandemic to reveal that certain risk factors fluctuated, particularly with the emergence of Omicron BA.2 dominance. They report variables such as mask-wearing habits, foreign travel, household size, working status, and contact with children or persons over 70 showed variable relationships with infection risk. The concluded the relevance of these factors may be linked to reinfection risk, variant infectiousness, and the status of social distancing regulations, emphasizing the dynamic nature of infection risks during different phases of the pandemic. Overall, the findings may seem interesting, but some questions can be raised as of their value now that the pandemic has largely subsided.

Major comments:

It is unclear to me why the authors did not include some of the statistically significant predictor variables that they reported in Table 1 (such as Traveling abroad, Rurality, and Employment status) in their subsequent analysis.

Minor comments:

Page 3: “We report a study to determine whether risk factors for infection after the emergence of the

omicron variant.”. Please rephrase for clarity.

Please replace “covid” with “COVID-19” across the manuscript when discussing and referring to COVID-19.

Table 1: Please highlight statistically significant values using the asterisk sign and define the p-value cutoff in the table's footnote.

6. PLOS authors have the option to publish the peer review history of their article (what does this mean?). If published, this will include your full peer review and any attached files.

Reviewer #1: No

Reviewer #2: No

---

## [Author Response · Author response to Decision Letter 0]

29 Jan 2024

A separate response to referees document has been supplied with submission.

---

## [Decision Letter · Decision Letter 1]

16 Feb 2024

Changing risk factors for developing SARS-CoV-2 infection from Delta to Omicron

PONE-D-23-35350R1

Dear Dr. Brainard,

We’re pleased to inform you that your manuscript has been judged scientifically suitable for publication and will be formally accepted for publication once it meets all outstanding technical requirements.

Kind regards,

Marwan Osman

Academic Editor

PLOS ONE

Reviewers' comments:

Reviewer's Responses to Questions

**Comments to the Author**

1. If the authors have adequately addressed your comments raised in a previous round of review and you feel that this manuscript is now acceptable for publication, you may indicate that here to bypass the “Comments to the Author” section, enter your conflict of interest statement in the “Confidential to Editor” section, and submit your "Accept" recommendation.

Reviewer #1: All comments have been addressed

Reviewer #2: (No Response)

2. Is the manuscript technically sound, and do the data support the conclusions?

Reviewer #1: Yes

Reviewer #2: (No Response)

3. Has the statistical analysis been performed appropriately and rigorously? 

Reviewer #1: Yes

Reviewer #2: (No Response)

4. Have the authors made all data underlying the findings in their manuscript fully available?

Reviewer #1: Yes

Reviewer #2: (No Response)

5. Is the manuscript presented in an intelligible fashion and written in standard English?

Reviewer #1: Yes

Reviewer #2: (No Response)

6. Review Comments to the Author

Reviewer #1: The authors addressed all comments reasonably and made corresponding revisions in the manuscript.

Reviewer #2: (No Response)

7. PLOS authors have the option to publish the peer review history of their article (what does this mean?). If published, this will include your full peer review and any attached files.

Reviewer #1: No

Reviewer #2: No
